# Applications of Radiomics and Radiogenomics in High-Grade Gliomas in the Era of Precision Medicine

**DOI:** 10.3390/cancers13235921

**Published:** 2021-11-25

**Authors:** Anahita Fathi Kazerooni, Stephen J. Bagley, Hamed Akbari, Sanjay Saxena, Sina Bagheri, Jun Guo, Sanjeev Chawla, Ali Nabavizadeh, Suyash Mohan, Spyridon Bakas, Christos Davatzikos, MacLean P. Nasrallah

**Affiliations:** 1Center for Biomedical Image Computing and Analytics (CBICA), University of Pennsylvania, Philadelphia, PA 19104, USA; Anahita.Kazerooni@pennmedicine.upenn.edu (A.F.K.); Hamed.Akbari@pennmedicine.upenn.edu (H.A.); Sanjay.Saxena@Pennmedicine.upenn.edu (S.S.); Jun.Guo@Pennmedicine.upenn.edu (J.G.); Ali.Nabavizadeh@pennmedicine.upenn.edu (A.N.); Suyash.Mohan@pennmedicine.upenn.edu (S.M.); Spyridon.Bakas@pennmedicine.upenn.edu (S.B.); Christos.Davatzikos@pennmedicine.upenn.edu (C.D.); 2Department of Radiology, Perelman School of Medicine, University of Pennsylvania, Philadelphia, PA 19104, USA; Sina.Bagheri@Pennmedicine.upenn.edu (S.B.); Sanjeev.Chawla@pennmedicine.upenn.edu (S.C.); 3Abramson Cancer Center, Perelman School of Medicine, University of Pennsylvania, Philadelphia, PA 19104, USA; sbagley@pennmedicine.upenn.edu; 4Glioblastoma Translational Center of Excellence, Abramson Cancer Center, University of Pennsylvania, Philadelphia, PA 19104, USA; 5Department of Pathology & Laboratory Medicine, Perelman School of Medicine, University of Pennsylvania, Philadelphia, PA 19104, USA

**Keywords:** GBM, radiomics, radiogenomics, imaging

## Abstract

**Simple Summary:**

Radiomics and radiogenomics offer new insight into high-grade glioma biology, as well as into glioma behavior in response to standard therapies. In this article, we provide neuro-oncology, neuropathology, and computational perspectives on the role of radiomics in providing more accurate diagnoses, prognostication, and surveillance of patients with high-grade glioma, and on the potential application of radiomics in clinical practice, with the overarching goal of advancing precision medicine for optimal patient care.

**Abstract:**

Machine learning (ML) integrated with medical imaging has introduced new perspectives in precision diagnostics of high-grade gliomas, through radiomics and radiogenomics. This has raised hopes for characterizing noninvasive and in vivo biomarkers for prediction of patient survival, tumor recurrence, and genomics and therefore encouraging treatments tailored to individualized needs. Characterization of tumor infiltration based on pre-operative multi-parametric magnetic resonance imaging (MP-MRI) scans may allow prediction of the loci of future tumor recurrence and thereby aid in planning the course of treatment for the patients, such as optimizing the extent of resection and the dose and target area of radiation. Imaging signatures of tumor genomics can help in identifying the patients who benefit from certain targeted therapies. Specifying molecular properties of gliomas and prediction of their changes over time and with treatment would allow optimization of treatment. In this article, we provide neuro-oncology, neuropathology, and computational perspectives on the promise of radiomics and radiogenomics for allowing personalized treatments of patients with gliomas and discuss the challenges and limitations of these methods in multi-institutional clinical trials and suggestions to mitigate the issues and the future directions.

## 1. Introduction

Glioblastoma (GBM) is the most common and malignant primary brain tumor, characterized by abundant proliferation of tumor cells, extensive infiltration in the surrounding brain parenchyma, genomic instability, robust angiogenesis, and resistance to therapies [1,2,3,4]. Built on the 2016 update of the World Health Organization (WHO) classification of CNS tumors [5], the 2021 fifth edition advances integration of molecular diagnostics with histopathological evaluation of brain tumors, including separating the previously designated entity of “glioblastoma” into IDH-wildtype glioblastoma and IDH-mutant grade 4 astrocytoma [6]. Clinical neuroimaging, along with histomolecular evaluation of tumor samples, helps in diagnosis and treatment planning of gliomas by capturing a vast amount of information about the tissue [7]. However, clinical images are mainly evaluated with a qualitative approach, and additional untapped potential remains, which may lead to a comprehensive picture of the tumor characteristics [8]. As radiologic scans reflect the underlying pathophysiology of the tumors, quantitative assessment of these images and development of imaging biomarkers with “radiomics” can aid in understanding the tumor biology [9,10,11] and treatment response [8,12,13,14]. Radiomics refers to extraction of high-throughput quantitative and mineable features characterizing the underlying pathophysiology of the tumor from medical images [12]. These computational features are synthesized via machine learning (ML) methods for prediction of an outcome [12,13,14]. With the rapid growth of computational algorithms, radiomics is now increasingly being applied to conventional and advanced neuro-oncologic imaging data to detect infiltrating margins of glial tumors, to differentiate treatment-related changes from true tumor progression, and to predict tumor infiltration, risk of future recurrence, and overall survival [8,12,13,14].

These radiomic and radiogenomic tools provide a noninvasive sampling of tumor microenvironments, the so-called “virtual biopsy”, allowing for a comprehensive evaluation of regional heterogeneity of these CNS tumors [15]. By providing these in vivo markers of spatial and molecular heterogeneity, these radiomic and radiogenomic tools have the potential to stratify patients into more defined diagnostic and therapeutic pathways and enable ”real-time” treatment surveillance in this era of personalized medicine [16].

Nevertheless, in clinical decision-making and patient care, a fully multi-modal and integrated diagnostic/prognostic approach incorporating radiomics, histology, and molecular data to provide a comprehensive picture of the tumor biology and evolution has not been broadly considered. This paper aims to provide and discuss the imaging scientist, neuro-oncology, and neuropathology perspectives regarding the current and potential roles of radiomics in precision medicine.

## 2. Can Radiomics Aid in Clinical Decision-Making? A Neuro-Oncology Perspective

The appeal of radiomics and radiogenomics in the care of patients with high-grade gliomas is obvious when one considers the primary diagnostic dilemmas faced in the neuro-oncology clinic, which begin the moment a patient presents to medical attention and continue through each line of treatment. Due to the marked interpatient heterogeneity of the disease in terms of both underlying tumor biology and clinical outcomes [4,17,18], one of the earliest challenges in the care of a patient with glioma is predicting the aggressiveness of that individual’s tumor and, therefore, the expected time to tumor recurrence and death. For the nearly 90% of patients with high-grade glioma who do not participate in a first-line clinical trial [19], temozolomide-based chemoradiotherapy is administered routinely despite great uncertainty around the degree of benefit that any one individual will gain from it. *MGMT* promoter methylation, the only biomarker used in routine practice to predict prognosis and benefit from temozolomide [20], has considerable interinstitutional variation in methodology and interpretation, limiting its true clinical utility [21,22]. This inability to reliably predict tumor behavior and clinical outcomes makes it difficult to efficiently risk-stratify patients for clinical trials and limits initial discussions with patients around expectations and goals of care.

Once standard chemoradiotherapy has been administered, assessing the tumor’s response to treatment is problematic, as conventional analysis of routine MRI cannot reliably distinguish between tumor progression and treatment-induced changes that can mimic tumor progression [23,24]; the latter, often referred to as pseudo-progression, may actually reflect efficacious treatment. This ambiguity leads to significant delays in clinical decision-making, causes undue anxiety in patients, and confounds efficacy assessment in clinical trials. Finally, when a patient’s tumor inevitably recurs, invasive neurosurgery for tissue acquisition is often needed to determine the tumor’s current molecular profile, which can differ substantially from the profile obtained at the time of initial diagnosis [25]. Even when repeat surgery is performed, however, there remains confusion about the clinical relevance of the molecular findings, as tumor sequencing is typically performed on only a single fragment of tissue. Because of the significant intratumoral heterogeneity that is a hallmark of high-grade glioma and occurs down to the single-cell level [26], it is unclear whether targeting a molecular alteration detected on routine sequencing will benefit the patient, as the alteration may not be present in the entirety of the tumor. Conversely, key molecular drivers of tumor growth may not be detected when only one region of the tumor is sequenced.

Once clinically validated and more easily implementable in routine practice, radiomics stands to solve or at least alleviate some of the aforementioned clinical challenges. Current radiomic methods may allow for accurate prognostication in newly diagnosed high-grade glioma [27]. The ability to predict clinical outcomes with a reasonable degree of confidence would offer significant value in the routine clinical care of patients with high-grade glioma and improve our conduct of high-grade glioma clinical trials. Although this information could be useful to some extent in all patients, one of the most exciting potential applications of radiomics for prognostication is the ability to predict a tumor’s aggressiveness and the patient’s clinical course prior to initial surgical intervention and tissue-based diagnosis of high-grade glioma. Accurate, noninvasive prognostication performed at the very start of a patient’s disease course, i.e., when a patient presents with radiographic findings highly concerning for high-grade glioma but has not yet undergone surgery for histopathologic confirmation and diagnosis of tumor type, could directly impact routine clinical decisions and usher in a new era of clinical trial design. For example, an octogenarian with a presumed glioblastoma based on MRI might elect to proceed with maximal safe resection followed by chemoradiotherapy if there is a high level of confidence that the patient will have a meaningful progression-free interval and an overall survival time over two years. Conversely, that same patient may elect for best supportive care only if it is expected that he or she will only live 6 or fewer months despite enduring a morbid surgical procedure followed by the side effects of chemoradiotherapy. In the context of clinical trial protocols, one can imagine a future where patients are immediately triaged to an experimental protocol rather than standard of care based on a radiomic signature predictive of poor outcomes with standard of care. This could happen prior to initial surgery, allowing the patient to receive experimental therapy either as a neoadjuvant approach or completely in lieu of standard surgical intervention, or following initial surgery as a way to prioritize patients for experimental alternatives to standard chemoradiotherapy and/or more efficiently stratify clinical trials based on prognosis.

Radiomics also has a potential future role in monitoring glioma response to therapy. Follow-up imaging of glioma patients commonly demonstrates new or increasing areas of enhancement in and around the resection bed which are concerning for tumor progression. However, in 20–30% of patients, this enhancement primarily represents treatment-related changes or pseudo-progression (PsP). Therefore, distinguishing treatment-related changes from true tumor progression (TP) is a common challenge and has critical implications in clinical decision-making [28]. Unfortunately, conventional imaging as well as existing response assessment criteria are limited in distinguishing treatment-related changes and tumor progression [29]. Radiomic methods may substantially increase our confidence in interpreting the changes observed on MRI following chemoradiotherapy, allowing for better clinical decision-making in routine practice and more accurate assessment of the efficacy of experimental therapies in clinical trials. The most significant challenge in validating radiomics-based tumor response assessment, and thus translating it for routine clinical use, has been the lack of a true gold standard for distinguishing between TP and PsP [25]. Serial imaging is difficult to use as a gold standard, since both tumor progression and pseudo-progression can worsen over time on repeat scans. Histopathologic confirmation is equally problematic, as histopathologic examination of chemoradiotherapy-treated high-grade glioma specimens has not been rigorously standardized. Material obtained from glioma re-resection typically contains a mixture of viable tumor, necrotic debris, and non-neoplastic brain elements with reactive changes [25]. Thus, even when tissue is acquired, most cases cannot be neatly dichotomized as “tumor progression” or “pseudo-progression”. Efforts are currently ongoing toward developing standard criteria for the pathological and molecular characterization of recurrent GBM [25], the acceptance of which would allow for validation of radiomics-based response evaluation. If these issues are resolved in the future, it is also possible that radiomic assessment of neuroimaging modalities other than MRI, such as amino acid PET, may lead to further improvements in disease monitoring.

Lastly, radiogenomics may also eventually change practice in the care of patients with glioma. The ability to detect clinically relevant tumor somatic mutations or copy number alterations noninvasively would have a positive impact on both routine management as well as clinical trials. Imaging-based ascertainment of the status of key molecular alterations that are commonly found in glioma, including *IDH* mutational status, *MGMT* methylation status, *EGFR* copy number, and mutational status and others, could allow for molecular profiling and optimal clinical management of patients with truly inoperable tumors or with unacceptable surgical risk due to comorbidities or performance status. Such technology would also allow for noninvasive molecular profiling at the time of tumor recurrence following frontline therapy. In clinical trials, one can envision a scenario where patients are screened for a neoadjuvant or window-of-opportunity study of a novel molecularly targeted therapy based on the presence of the therapeutic target as determined by radiogenomics. This type of trial design could revolutionize the way we evaluate early phase therapeutic candidates by allowing for tissue-based pharmacodynamic assessment of novel drugs in the newly diagnosed setting.

## 3. What Radiomics Offers: A Computational Perspective

Advances in measurement methods, including medical imaging and genomic sequencing of the tumor, have enormously increased the amount of patient data available to the clinician. The amount and breadth of the data have reached an overwhelming point; interpretation requires specialization, and complete integration is not always possible. Radiomics methods can play a critical role in objectively and reproducibly recognizing and quantifying the underlying complex patterns in the data that are not discernible by humans, thereby complementing and supporting the clinical decision-making regarding the best course of treatment for glioma patients [30,31]. Radiomics has aided in characterization of the tumors through segmentation, diagnosis, prognosis, and subtyping as well as played a role in monitoring gliomas and their responses to treatment (Figure 1). While the prognosis prediction does not yet have implications for upfront therapy in routine clinical practice, improved prognostication may impact clinical trial eligibility and has potential for precision and personalized treatment planning [32,33].

Imaging characteristics of infiltration, cellularity, microvascularity, spatial location of tumor, volume of compartments, morphology, and blood–brain barrier compromise integrated via radiomic analysis have the potential to reveal imaging patterns that are highly predictive of clinical outcome and prognosis of patients, as documented in several studies [27,33,34,35,42]. Furthermore, advanced computational analytics via radiomics for evaluation of response to treatment and distinguishing TP from PsP have provided rich and highly informative characterization of the tumor and its surrounding tissues, extending the evaluation of tissue properties beyond the capabilities of human visual interpretation [38,43,44,45]. Specifically, these studies demonstrate that patients with TP demonstrate imaging features reflecting higher angiogenesis, higher cellularity, lower necrosis, and lower water concentration [38].

Radiogenomics, or imaging genomics, has emerged as a powerful tool for discovery of molecular associates of radiographic phenotypes. To accomplish this, a number of unique approaches have been developed. Most of the existing radiogenomics studies have adopted an exploratory analysis approach to investigate the relationships between molecular dynamics and tumor characteristics reflected by specific radiographic phenotypes (radiophenotypes) [11]. For example, radiophenotypes, including tumor enhancement, nonenhancing tumor, necrosis, infiltrated edema, neo-angiogenesis, microstructural changes, and tumor location, have been associated with genomic profiles of the tumors to provide a better understanding of the underlying tumor biology [11,46,47,48,49,50,51,52,53]. Along these lines, a few radiogenomics studies have stratified high-grade glioma patients based on their risk, i.e., into groups of high, intermediate, and low-risk based on radiomic features that were predictive of overall or progression-free survival, and explored associations of these predictive radiomic features with gene expression profiles [54].

In contrast to exploratory studies, hypothesis-driven radiogenomic approaches in the literature have specified a radiogenomic signature that aims to provide upfront prediction of genetic mutations or expression levels in the patients based on their pre-operative MRI scans [11]. The multi-pronged goal of this method is to overcome the sampling errors that occur with biopsies, to guide personalized therapies, and to encourage development of future targeted drug therapies [11]. Therefore, the focus has been on development of imaging signatures for mutations in several driver genes, including *IDH*, *EGFR* (including *EGFRvIII* mutation), *PTEN*, and *TP53*; for key pathways (*RTK*, *PI3K*, *MAPK*, etc.) and molecular subtypes; as well as for *MGMT* promoter methylation status [36,37,55,56,57,58,59].

The problem is also being approached from another perspective. Subgroups of similar patients are identified based on their imaging phenotypes, with the goal of understanding how to customize therapies for individuals [60]. Dramatic inter-tumoral heterogeneity from patient to patient exists as a result of expression of specific molecular markers or response to treatments [61,62]. In a few studies, distinct imaging subtypes of high-grade gliomas have been discovered and proven to correlate with molecular subtypes and overall survival in patients beyond *IDH* status [61]. With rapid development of radiomics, semi-supervised learning methods [63,64] or multi-modal learning based on multi-omic data (e.g., genomic, transcriptomic, radiomic) are areas of promise to noninvasively characterize subtypes of high-grade gliomas and power clinical trials by facilitating patient stratification [65].

Within individual patients’ tumors, intra-tumoral heterogeneity is a contributor to treatment failure in glioblastoma, due to diversity of genetic and transcriptomic aberrations across the tumor landscape, which can lead to resistance of the tumor to standard of care therapies and rapid recurrence [66]. This heterogeneity may not be detected by histomolecular examination of tumor samples collected during biopsy or portions of the tumor obtained during limited surgical resection [11,67]. This challenge to patient care is the perfect opportunity for radiomics and radiogenomics to provide noninvasive assessment of glioblastoma heterogeneity prior to treatment through in vivo biomarkers [68]. For example, the method of habitat imaging identifies distinct functional tumorous regions and cell populations based on image characteristics determined through radiomics and machine learning, allowing a comparison among these subregions based on radiographic imaging and histologic findings [8,12,69].

These techniques may be made more powerful through incorporation of data from advanced imaging modalities with conventional scans in radiomics studies. Complementing structural features with functional and biological characteristics of the tumor in this way provides a more comprehensive picture of tumor evolution and response to treatment. For instance, radiomics features extracted from structural MRI sequences, DWI, susceptibility-weighted imaging (SWI), 55-direction high angular resolution diffusion imaging (HARDI), and arterial spin labelling (ASL) were able to predict *IDH* and *ATRX* mutations and chromosome 7/10 aneuploidies with high accuracy and *CDKN2* family mutations with relatively high accuracy [70]. Another example of an advanced MRI technique is amine CEST echoplanar imaging (CEST-EPI), which is a fast MRI molecular imaging sequence to measure tumor pH [71]. This technique has been shown to be effective as a noninvasive biomarker to determine *IDH* mutational status and 1p/19q co-deletion status, as well as being of value as an early imaging biomarker for bevacizumab treatment response and failure in recurrent glioma [71,72]. Recently, it has been shown that principal component analysis of DSC MR perfusion images combined with support vector machine (SVR) approaches had moderate to strong correlation with CEST-EPI PH maps [40].

Moving one step further than radiogenomics, radio-patho-genomic analyses incorporate microscopic tissue-scale imaging with radiographic scans and molecular information of the tumor tissue. Advanced computational analysis of this powerful data combination may provide new clinical insights into tumor biology and the involved pathological processes to further aid personalized diagnostics and precision medicine [36,37,59,73,74,75].

Separate but likely related to the molecular features of glioma is the tumor microenvironment. Composed of tumor cells, vascular endothelial cells, stromal cells, astrocytes, microglia, immune cells, extracellular matrix proteins, and cytokines, the microenvironment plays a critical role in tumor cell invasion, resistance to treatments, recurrence, and as a result, poor patient prognosis. Unraveling this complex microenvironment could be helpful in tumor prognostication and treatment planning [76,77]. The literature regarding use of radiomics to predict the tumor microenvironment is sparse. In a recent study, tumor radiomics signatures derived from apparent diffusion coefficient maps was able to predict the tumor immune phenotypes, including T cell fraction (enriched vs. deficient group), T cell subclass fraction, and tumor-associated macrophage (TAM) fraction [78]. Another study discovered a radiomic subtype of GBM with poor prognosis which might respond better to immunotherapy [79]. These findings might help answer critical clinical concerns, including determining the difference between TP and PsP or radiation necrosis [80]. Given the importance of tumor microenvironment, this is an emerging field in radiomics, and many studies are currently underway.

Over the past few years, several challenges with the clinical utility of radiomics in diagnosis and treatment of high-grade gliomas have been highlighted, and an increasing number of studies have attempted to address these concerns. One of the main challenges to the widespread clinical implementation of radiomic approaches is to demonstrate their generalizability. Variations in image acquisition protocols between sites and even across scanners at a single site secondary to differences in image contrast, voxel resolutions, slice thicknesses, image reconstruction methods, magnetic field strengths, echo, and repetition times can be an impediment to reproducibility of results in radiomic studies. These concerns have led to standardization of imaging efforts for multicenter neuro-oncology trials. The first consensus recommendation was published in 2015 with a pragmatic approach of developing a balanced protocol that would be feasible for most centers and could reach large-scale compliance and acceptance from the community and would be applicable to both 3 and 1.5 T scanners [81]. Key elements of the suggested protocol included pre and post-contrast volumetric, inversion recovery prepared t1-weighted gradient echo MRI, an axial, 2-dimensional FLAIR sequence with a turbo-spin-echo (TSE) readout; an axial, 2-dimensional, 3-directional (isotropic) diffusion-weighted imaging (DWI) sequence obtained using echoplanar or radial acquisition; and an axial, 2-dimensional T2-weighted TSE sequence [82]. More recently, consensus recommendations for dynamic susceptibility contrast (DSC) MRI protocol for use in high-grade gliomas were published [82] and concluded that full-dose preload and full-dose bolus dosing using intermediate flip angle and field strength-tailored time to echo (40–50 ms at 1.5 T, 20–35 ms at 3 T) provides overall best accuracy and precision for cerebral blood volume estimates [82]. It has also been suggested that in situations where such double dose bolus injection is not desirable, no-preload, full-dose bolus dosing with a low flip angle (30°) and a suitable time to echo provides comparable performance and accuracy [82].

Another barrier to the development and broader adoption of accurate and translatable computational algorithms in radiomics analysis of gliomas is the need for ample training datasets with accurate corresponding labels. Many modern medical datasets include clinical data in multiple modalities. Often, comprehensive datasets span from patient background to radiology scans, to histology sections, to genetic assays. Due to the cost of data storage steadily decreasing over time, it becomes increasingly more accessible for institutions large and small to create publicly accessible datasets. Additionally, advances in lab methods, imaging, and storage mean that the stored data can have a higher resolution, depth, and fidelity. Large biomedical repositories such as The Cancer Imaging Archive (TCIA, www.cancerimagingarchive.net, access date: 19 october 2021) [83] have proven useful in expediting scientific discovery. TCIA has streamlined the process of reproducibility analyses through the release of “Analysis Results”. Publications in these ”Analysis Results” describe augmentations of the TCIA data collections through newly released expect annotations [84,85,86], as well as further data analysis such as outcome prediction and exploratory radiogenomic analysis [42,47,59,87]. These large-scale repositories and analysis results can serve as exemplary resources for single centers to create their databases in a systematic approach. Furthermore, generalizability of the developed methods can be examined by using these freely available datasets as a discovery cohort for ML model training which will be subsequently tested independently on an institutional patient cohort. To facilitate its clinical translation, radiomics can be incorporated with the Picture Archiving and Communication System (PACS). Radiomic data can be stored alongside the DICOM metadata and images and facilitate future statistical and predictive modeling. Such integration supports creation of an institutional database in a standardized and harmonized approach [88].

In medical image analysis literature, there have been numerous manuscripts introducing novel methods always superior to the previously published methods. However, when looking closer there seems to be an unfair comparison across published methods, as they have been evaluated on different datasets. It was predominantly the need for a common benchmarking environment and dataset that spearheaded the birth of computational competitions, also known as challenges, where computational researchers can develop and compare their methods fairly. As an example, here we refer to the landmark international challenge on brain tumor segmentation (BraTS) [85,86,89,90,91], in conjunction with the conference on medical image computing and computer-assisted interventions (MICCAI), that has been leading the development of brain tumor segmentation algorithms.

Finally, consortiums play a key role in supporting the collection of large and diverse data for learning the underlying patterns of the diseases and overcoming the so-called “curse of dimensionality” problem [16,92]. As an example, the Radiomics Signatures for Precision Diagnostics (ReSPOND) consortium, as an international initiative for machine learning in glioma imaging [16], was formed for further development, generalization, and clinical translation of radiomic-based biomarkers for personalized prognostication. ReSPOND is a collaborative effort of approximately 20 international institutions across the globe. With diverse and extensive data, this consortium aims to investigate biomarkers for prediction of risk in terms of overall and progression-free survival, upfront prediction of tumor recurrence, differentiation of TP from PsP, and prediction of molecular characteristics of gliomas [16,93]. Through such consortiums, multiple aspects of radiomic analysis reproducibility can be tackled, and guidelines be provided for the research community. As an example, the impact of image preprocessing techniques on reproducibility of radiomic feature computations, as regulated by the Image Biomarker Standardization Initiative (IBSI), and the potential of image harmonization techniques to mitigate the variability of MRI scans [94,95,96,97,98] can be evaluated in a single controlled setting on a diverse cohort of data.

## 4. Should Radiomics Be Integrated with WHO Classification? A Neuropathology Perspective

Classification of gliomas continually evolves with the ongoing study of tumor biology and clinical courses and outcomes. Changes in diagnostic criteria have been dramatic over the past five years, with the advent of the 2016 update to the WHO classification [5], as well as the establishment of and publications from The Consortium to Inform Molecular and Practical Approaches to CNS Tumor Taxonomy—Not Official WHO (cIMPACT-NOW [99]), and now the 2021 WHO classification [6]. Gliomas previously belonging to few general categories are now parsed into many different tumor types based on a combination of characteristics, including molecular features, that make the different tumors unique entities and that correlate with clinical course.

The field of surgical pathology involves specimen analysis at both a gross and a microscopic level, with the two different scales giving complementary and equally crucial information. However, in neuropathology, the gross assessment of glioma surgical specimens can be limited due to the inherent characteristics of glial tissue and surgical methods. Fortunately, correlation with radiological studies often can serve as a surrogate for tissue gross examination, and this correlation is considered by many neuropathologists to be a critical element of the diagnostic process. Therefore, it is natural that as more information is derived from radiological images through machine learning, these results may be added to the list of relevant properties assessed to completely characterize a glioma.

Noninvasive radiogenomic classification of tumors into categories defined by the WHO [100,101,102] as well as radiomic prediction of *MGMT* promoter methylation status [103] have clear benefits in assessing tumors pre-operatively and through a patient’s course of care. These predictions will allow administration of neoadjuvant or intraoperative targeted therapy when these become available, and they will allow complete assessment of tumors that have limited tissue available for laboratory testing, when surgeries yield sparse tissue at resection or when the tumors are not resectable due to location or patient clinical status. In addition, radiogenomic studies have the potential to assess molecular heterogeneity and provide more information than that gleaned from laboratory testing on a single fragment of tissue. These benefits may eventually be realized at low cost in terms of time and money, compared to expensive molecular testing with relatively long turnaround times.

In this paper, the radiomic feature of tumor location is mentioned several times. Location has become important in the classification of several brain tumor types, such as diffuse midline gliomas and ependymomas. For the former, the diagnosis cannot be made without tumor involvement of midline brain structures. For the latter, ependymoma classification is first stratified by location in the spine, posterior fossa, or supratentorial compartment. Depending upon location, different molecular features must be interrogated to come to a final diagnosis. Although making the diagnosis of a high-grade glioma such as an IDH-wildtype glioblastoma or an IDH-mutant grade 4 astrocytoma does not currently depend on location, alternative location-dependent diagnoses such as diffuse midline glioma must be ruled out. In addition, it is possible that as molecular features and prognoses continue to be studied, additional subsets of glioblastoma and IDH-mutant astrocytoma will be teased out that correlated with location. This location information and its integration with other features predictive of histology and molecular changes are natural applications of radiomics.

Machine learning on radiologic imaging has additional potential. One of the most interesting categories of brain tumors is designated “not elsewhere classified (NEC)”. A “NEC” tumor has been tested for known relevant molecular features, but despite the availability of results, the tumor cannot be classified as one of the currently understood entities [6]. Radiomics and radiogenomics may lend biological and clinical understanding to these novel entities through prediction of survival and suggestion of involved biological signaling pathways associated with low-risk and high-risk categories [50,104,105]. The utility of the machine learning results may be envisioned in multiple ways. For a glioma that is classified at “NEC” with no understood molecular features discovered by molecular testing, the imaging features may suggest the pathways relevant to treatment and/or prognosis. Alternatively, for a glioma classified as “NEC” with a combined molecular and histologic profile that does not lead to clear understanding of the patient’s prognosis, the imaging features and machine learning outcome prognostic prediction may shed light on the clinical course and hence guide treatment planning. This understanding will allow clinical care to be tailored to the patient. In addition, if biological pathways are implicated by the imaging features, these will provide the basis for new directions of basic science investigation into gliomagenesis and progression.

Ultimately, with progress in the field, the potential exists for radiomic and radiogenomic criteria to be utilized as biomarkers in tumor characterization, in the way that histologic and molecular features are currently used in the WHO classification [6]. These imaging criteria may be incorporated into diagnoses as molecular and histologic features are currently enumerated and integrated in layered diagnoses [106] (Figure 2).

## 5. Conclusions

Radiomics in medical imaging analysis of gliomas has introduced novel solutions to the current clinical challenges for treatment of gliomas and has shown promising evidence for personalized diagnosis and treatments. Key applications of radiomics and radiogenomics include risk stratification of glioma patients by upfront projection of the OS and PFS, prediction of spatial location of tumor recurrence, distinguishing TP from PsP, and prediction of the molecular properties of the tumor and spatial heterogeneity. As discussed throughout this paper, radiomics has the potential to support management of gliomas but is not yet translated into clinical decision-making. Current radiomics efforts may benefit from addressing the challenges of reproducibility and generalizability and exploring the impact of fully integrated diagnostics in clinical management of glioma patients.

## Figures and Tables

**Figure 1 cancers-13-05921-f001:**
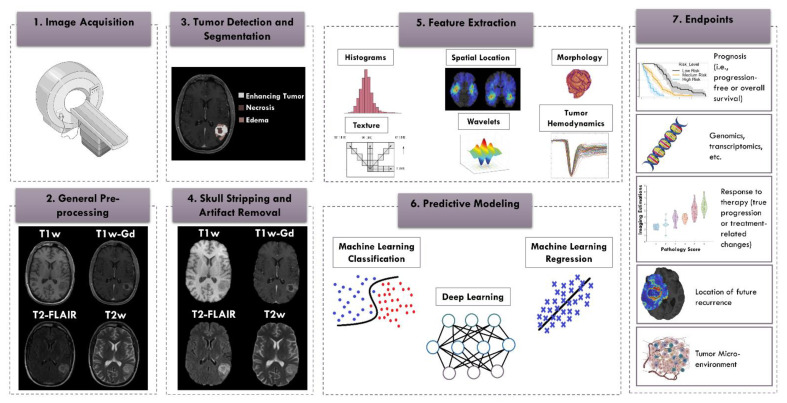
Radiomics pipeline. From left to right: (**1**) image acquisition; (**2**) general image pre-processing including image re-orientation, co-registration of the images, and alignment of images with a reference atlas; (**3**) tumor detection and segmentation; (**4**) skull stripping and artifact removal (bias field, noise, etc.); (**5**) feature extraction, such as features of histogram, texture, wavelets, location, morphology, and hemodynamics; (**6**) predictive modeling using classification or regression; (**7**) prediction of endpoints, such as patient’s survival [34,35], genomics [11,36,37], response to therapy [38], site of future recurrence [39], or tumor micro-environment [40] (Some graphics are from Servier Medical Art: smart.servier.com (access date: 14 January 2021) and [41]).

**Figure 2 cancers-13-05921-f002:**
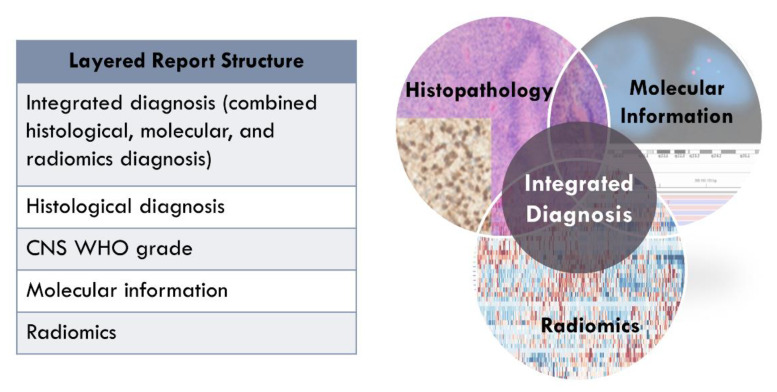
The hope: integrating radiomics into the layered diagnosis.

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
