# Peer review of "Applications of Radiomics and Radiogenomics in High-Grade Gliomas in the Era of Precision Medicine"

_cancers, 2021, doi:10.3390/cancers13235921_

Round 1

Reviewer 1 Report

This review presents an expert perspective on applications of ML in image analysis of gliomas for personalized approaches to patient care. The authors emphasize the untapped potential that exists in image analysis and how critical it is to identify methods for translation of ML into clinical practice. One of the important areas of application of ML is in characterization of tumor heterogeneity and targeted tissue biopsy and genetic analysis of different regions of the tumor. This is a well written review and presents a great overview of future of precision medicine. I recommend only minor modifications to text.

minor modifications:

line 48-49: 'treatment related changes', not 'changed'

line 61-61: from the point of view of imaging scientist

line 303: please include the DSC recommendations for no bolus protocol and low flip angle. Many institutions don't use double dose gadolinium because of concerns for gadolinium deposition in the brain and imaging of patients with renal failure

line 330: in this paragraph there is a great description of the RESPOND consortium. It would be great to include a table in this section that describes the different methods of radiomic feature extraction and methods for regulation of radiomic feature extraction.  This will define the rigorous approach that is needed for feature extraction and a wide range of features that can be extracted.

- during discussion of incorporation of radiomics into WHO classification of tumors, it would be important to include tumor location being a very important part of WHO classification.  Tumor location is not currently clearly part of WHO classification and incorporating it would provide additional critical information.

- please discuss methods that will improve translation of radiomics into clinical practice - data sharing, development of databases by individual hospitals, analytic tools that are available to clinical radiologists, PACS based tools that allow database building and classification

Author Response

This review presents an expert perspective on applications of ML in image analysis of gliomas for personalized approaches to patient care. The authors emphasize the untapped potential that exists in image analysis and how critical it is to identify methods for translation of ML into clinical practice. One of the important areas of application of ML is in characterization of tumor heterogeneity and targeted tissue biopsy and genetic analysis of different regions of the tumor. This is a well written review and presents a great overview of future of precision medicine. I recommend only minor modifications to text.

minor modifications:

line 48-49: 'treatment related changes', not 'changed'

line 61-61: from the point of view of imaging scientist

Response: Thank you for catching these errors; we have corrected them.

line 303: please include the DSC recommendations for no bolus protocol and low flip angle. Many institutions don't use double dose gadolinium because of concerns for gadolinium deposition in the brain and imaging of patients with renal failure

Response: Thanks for your helpful comment. We have added it to the text (Lines 308-311):

“It has also been suggested that in situations where such double dose bolus injection is not desirable, no-preload, full-dose bolus doing with a low flip angle (30°) and a suitable time to echo provides comparable performance and accuracy [83].”

line 330: in this paragraph there is a great description of the RESPOND consortium. It would be great to include a table in this section that describes the different methods of radiomic feature extraction and methods for regulation of radiomic feature extraction. This will define the rigorous approach that is needed for feature extraction and a wide range of features that can be extracted.

Response: Thank you for your response. We agree with this comment that the methods for regulation of feature extraction should be mentioned but the detailed explanation is out of scope of this paper. To address this reviewer’s comment we have added the following to lines 356-362 of the manuscript:

“Through such consortiums, multiple aspects of radiomic analysis reproducibility can be tackled and guidelines be provided for the research community. As an example, the impact of image preprocessing techniques on reproducibility of radiomic feature computations, as regulated by the Image Biomarker Standardization Initiative (IBSI), and the potential of image harmonization techniques to mitigate the variability of MRI scans can be evaluated in a single controlled setting on a diverse cohort of data.”

- during discussion of incorporation of radiomics into WHO classification of tumors, it would be important to include tumor location being a very important part of WHO classification. Tumor location is not currently clearly part of WHO classification and incorporating it would provide additional critical information.

Response: Thank you for pointing out the importance of location in tumor classification. Indeed, as greater understanding is gained of certain tumor types, their classification has come to include location.  For example, ependymoma classification is first stratified by location in the spine, posterior fossa, or supratentorial compartment.  Depending upon location, different molecular features must be interrogated to come to a final diagnosis.  Although diagnoses of high-grade gliomas such as glioblastoma and IDH-mutant grade 4 astrocytomas do not currently depend on location, it is possible that as molecular features and prognoses are correlated with location, some importance for location will surface, just as diffuse midline glioma has been separated from other high-grade gliomas.  We have added in the ability of radiomics to discover and contribute this information to the final section as follows:

“In this paper, the radiomic feature of tumor location has been mentioned several times. Location has become important in the classification of several brain tumor types, such as diffuse midline gliomas and ependymomas. For the former, the diagnosis cannot be made without tumor involvement of midline brain structures. For the latter, ependy-moma classification is first stratified by location in the spine, posterior fossa, or supraten-torial compartment. Depending upon location, different molecular features must be inter-rogated to come to a final diagnosis. Although making the diagnosis of a high-grade gli-oma such as an IDH-wildtype glioblastoma or an IDH-mutant grade 4 astrocytoma does not currently depend on location, alternative location-dependent diagnoses such as dif-fuse midline glioma must be ruled out. In addition, it is possible that as molecular features and prognoses continue to be studied, additional subsets of glioblastoma and IDH-mutant astrocytoma will be teased out that correlated with location. This location information and its integration with other features predictive of histology and molecular changes are nat-ural applications of radiomics.”

- please discuss methods that will improve translation of radiomics into clinical practice - data sharing, development of databases by individual hospitals, analytic tools that are available to clinical radiologists, PACS based tools that allow database building and classification

Response: Thanks for your helpful note. We have added the following passage to address this comment (Lines 326-334):

“These large-scale repositories and analysis results can serve as exemplar resources for single centers to create their databases in a systematic approach. Furthermore, generalizability of the developed methods can be examined by using these freely available datasets as a discovery cohort for ML model training which will be subsequently tested independently on institutional patient cohort. To facilitate its clinical translation, ra-diomics can be incorporated with Picture Archiving and Communication System (PACS). Radiomic data can be stored alongside the DICOM metadata and images and facilitate future statistical and predictive modeling. Such integration supports creation of an institutional database in a standardized and harmonized approach.” 

Reviewer 2 Report

This manuscript gives an informative perspective of the expectations Neurooncologists and Neuropathologiest may have from the ongoing developments in the area of radiomics in GBMs.

A few thoughts and comments:

I think the introduction section should at least include a very brief definition of what radiomics actually is. Additionally, the terms AI and ML, or “AI-based radiomics” (line 52) are used throughout this manuscript. This can get confusing, without a proper definition. E.g. I do not understand the first sentence of the abstract. Do you define radiomics just a feature extraction step, or does it also include a feature selection step and modelling step? What is the difference to ML? Starting the conclusion (line 395) with “Artificial intelligence….” while the title speaks of radiomics, adds to the confusion.

Lines 45-48: Please give references for these claims, especially for the “precise” part. You rightly mentioned in your manuscript that gold standards are lacking, so how can we even make this claim? I see this as a big challenge in the field. 

Line 51: Ref. 10 is not a peer reviewed and therefore should not be cited here.

“as routine MRI cannot reliably distinguish between tumor progression and treatment-induced changes that can mimic tumor progression” (lines 80-81), yet in line 95-96 you state that radiomics can solve some of the challenges. I get it, “some” is the key word here. Still, care needs to be taken not to downplay the role of high quality input data (e.g. quantitative and functional MRI sequences) that give much more information than just anatomy.  Radiomics on routine MRI will only provide so much information (rubbish in = rubbish out).

References:

Refs 22, 25, 69 wrong or incomplete citations

Author Response

This manuscript gives an informative perspective of the expectations Neurooncologists and Neuropathologiest may have from the ongoing developments in the area of radiomics in GBMs.

A few thoughts and comments:

I think the introduction section should at least include a very brief definition of what radiomics actually is. Additionally, the terms AI and ML, or “AI-based radiomics” (line 52) are used throughout this manuscript. This can get confusing, without a proper definition. E.g. I do not understand the first sentence of the abstract. Do you define radiomics just a feature extraction step, or does it also include a feature selection step and modelling step? What is the difference to ML? Starting the conclusion (line 395) with “Artificial intelligence….” while the title speaks of radiomics, adds to the confusion.

Response: Thank you for pointing out this important clarification. We have modified the text and specifically artificial intelligence to keep consistency throughout the text and avoid any confusions. 

Lines 45-48: Please give references for these claims, especially for the “precise” part. You rightly mentioned in your manuscript that gold standards are lacking, so how can we even make this claim? I see this as a big challenge in the field. 

Response: This is important, thank you for pointing it out. We have modified the text to avoid overclaiming the results of radiomics, and added references to illustrate the work that has been done.

Line 51: Ref. 10 is not a peer reviewed and therefore should not be cited here.

Response: Reference 10 is a peer-reviewed article published in the Journal of Neurosurgery (Khanna, O.; Fathi Kazerooni, A.; Farrell, C.J.; Baldassari, M.P.; Alexander, T.D.; Karsy, M.; Greenberger, B.A.; Garcia, J.A.; Sako, C.; Evans, J.J.; et al. Machine Learning Using Multiparametric Magnetic Resonance Imaging Radiomic Feature Analysis to Predict Ki-67 in World Health Organization Grade I Meningiomas. Neurosurgery 2021, 89, 928–936, doi:10.1093/neuros/nyab307).

“as routine MRI cannot reliably distinguish between tumor progression and treatment-induced changes that can mimic tumor progression” (lines 80-81), yet in line 95-96 you state that radiomics can solve some of the challenges. I get it, “some” is the key word here. Still, care needs to be taken not to downplay the role of high quality input data (e.g. quantitative and functional MRI sequences) that give much more information than just anatomy. Radiomics on routine MRI will only provide so much information (rubbish in = rubbish out).

Response: Thank you for that important point.  We have modified the sentence to specify that conventional analysis of MRI scans cannot distinguish between tumor progression and treatment-induced changes; radiomics has greater power to detect differences by calculating features that can reveal the underlying pathophysiology of the tumor. We provide references in which conventional and advanced MRI techniques and ML are used to discriminate these disease entities from each other. 

References:

Refs 22, 25, 69 wrong or incomplete citations

Response: Thanks for your note; we have fixed the references.

Please let me know if any further changes should be made.  Thank you for your consideration of our manuscript.